# Diversity of HLA-A2-Restricted and Immunodominant Epitope Repertoire of Human T-Lymphotropic Virus Type 1 (HTLV-1) Tax Protein: Novel Insights among N-Terminal, Central and C-Terminal Regions

**DOI:** 10.3390/biom13030545

**Published:** 2023-03-16

**Authors:** Thaiza Aline Pereira-Santos, Anderson Santos da Rocha, Ágata Lopes-Ribeiro, Laura Cardoso Corrêa-Dias, Patrícia Melo-Oliveira, Erik Vinicius de Sousa Reis, Flávio Guimarães da Fonseca, Edel Figueiredo Barbosa-Stancioli, Moriya Tsuji, Jordana Grazziela Alves Coelho-dos-Reis

**Affiliations:** 1Laboratório de Virologia Básica e Aplicada (LVBA), Departamento de Microbiologia, Instituto de Ciências Biológicas, Universidade Federal de Minas Gerais, Belo Horizonte 31270-901, MG, Brazil; 2Centro de Tecnologia em Vacinas (CT-Vacinas), Parque Tecnológico de Belo Horizonte, Belo Horizonte 31310-260, MG, Brazil; 3Aaron Diamond AIDS Research Center, Division of Infectious Disease, Department of Medicine, Columbia University Irving Medical Center, New York, NY 10032, USA

**Keywords:** HTLV-1, Tax, MHC-I peptides, CD8^+^ T-cell response, HLA-A2 haplotype, immunoinformatics, immunodominance

## Abstract

The present study sought to search for the immunodominance related to the N-terminal, Central and C-terminal regions of HTLV-1 Tax using novel, cutting-edge peptide microarray analysis. In addition, in silico predictions were performed to verify the presence of nine amino acid peptides present along Tax restricted to the human leukocyte antigen (HLA)-A2.02*01 haplotype, as well as to verify the ability to induce pro-inflammatory and regulatory cytokines, such as IFN-γ and IL-4, respectively. Our results indicated abundant dose-dependent reactivity for HLA-A*02:01 in all regions (N-terminal, Central and C-terminal), but with specific hotspots. Furthermore, the results of fold-change over the Tax^11–19^ reactivity obtained at lower concentrations of HLA-A*02:01 reveal that peptides from the three regions contain sequences that react 100 times more than Tax^11–19^. On the other hand, Tax^11–19^ has similar or superior HLA-A*02:01 reactivity at higher concentrations of this haplotype. The in silico analysis showed a higher frequency of IFN-γ-inducing peptides in the N-terminal portion, while the C-terminal portion showed a higher frequency of IL-4 inducers. Taken together, these results shed light on the search for new Tax immunodominant epitopes, in addition to the canonic Tax^11–19^, for the rational design of immunomodulatory strategies for HTLV-1 chronic diseases.

## 1. Introduction

Human T-cell lymphotropic virus type 1 (HTLV-1) or primate T-lymphotropic virus type 1 (PTLV-1), according to the International Committee on Taxonomy of Viruses (ICTV), belongs to the *Retroviridae* family, *Deltaretrovirus* genus, and was the first human retrovirus described. HTLV was isolated from infected T cells of a patient diagnosed with cutaneous lymphoma [1]. HTLV-1 is the etiological agent of two major human diseases: HTLV-1-associated myelopathy/tropical spastic paraparesis (HAM/TSP), a pathology associated with chronic demyelinating neuroinflammation, and adult T-cell lymphoma/leukemia (ATLL) [2,3]. In addition, other inflammatory diseases of clinical importance are also correlated with HTLV-1 infection, such as uveitis, polymyositis, infectious dermatitis, arthropathies, Sjogren’s syndrome, among others [4,5,6].

HTLV-1 infection is widespread worldwide with a high prevalence of seropositive individuals in Japan, the Caribbean islands and Brazil, as well as countries in South America, Africa and the Middle East, and a low prevalence in regions of Austral-Melanesia. The worldwide estimate of HTLV chronic infection ranges from 15 to 20 million people, although this number could be underestimated [7,8]. Among these countries, Brazil stands out as the country with the highest absolute number of cases, whose estimate reaches 2 million carriers of HTLV-1 [9,10].

The HTLV-1 proviral genome, consisting of 9 kb, comprises the *gag*, *pol* and *env* gene regions flanked by long repeat terminations (LTRs) at the 3′ and 5′ ends [11]. A region called *pX* is found between the *env* gene and the LTR flanking domain at the 3′ end, which has four open reading frames (ORFs). ORF IV represents the Tax protein sequence [12], which codes for a 40 kDa immunogenic phosphoprotein important in different cell activation and signaling processes, as well as in the cell cycle [13]. Tax is strongly associated with the carcinogenic effect of HTLV-1, which is well demonstrated and characterized [12,14,15,16].

Tax is also involved in the activation and expansion of T cell clones infected by HTLV-1, which take part in the induction of a prominent and robust proinflammatory response in the host. Tax^11–19^ (LLFGYPVYV) epitope present in the N-terminal region of the protein has been associated with the induction of cytotoxic CD8^+^ T cells that respond by producing high levels of IFN-γ, which is more prominent in HAM/TSP patients [17]. On the other hand, the C-terminal end of Tax is associated with the development of Tax^301–309^-specific cytotoxic CD8^+^ T-cell responses, which were recently linked with protection against challenge with MT-2 cell lines using a humanized mouse model [18,19].

Although much is known about CD8^+^ T-cell responses in HTLV-1 infection, there are no vaccines or therapeutic strategies aimed at combating HTLV-1 to promote viral clearance by harnessing this knowledge. In this scenario, the development of immunotherapeutic or vaccine strategies becomes crucial for controlling the diseases associated with the virus. Considering the persistence of this retrovirus and that a humoral immune response would be insufficient for controlling the proviral load, vaccines harnessing the proinflammatory response of cytotoxic T lymphocytes (CTLs) induced by Tax could contribute as therapeutic strategies [19,20,21,22].

It is also important to consider the proinflammatory versus modulatory profile induced by immunodominant epitopes of Tax protein, ensuring a long-term protective response [20]. In this sense, the present work intended to understand the immunodominance epitope profile of Tax protein in its different regions (N-terminal, Central and C-terminal portions) using cutting-edge peptide microarray technology and immunoinformatic tools based on the binding of class I Major Histocompatibility Complex (MHC-I) molecule, HLA-A*02:01.

## 2. Materials and Methods

Intending to broaden the understanding of immunodominant peptides from Tax, in vitro and in silico assessment of peptide-HLA-A*02:01 interaction was evaluated. A comprehensive in silico analysis of the HLA-A*02:01-restricted peptide signature of N-terminal, Central and C-terminal regions of Tax was performed. For the in vitro analysis, cutting-edge PEPperPRINT © peptide microarray technology was employed in order to assess HLA-A*02:01 reactivity. A flowchart of the study is presented in Appendix A.

### 2.1. In Silico Analysis for Epitopes Prediction Restricted to HLA-A*02:01

Full Tax sequences (GenBank: AYN25353.1) of HTLV-1 were acquired from the NCBI protein database available at https://www.ncbi.nlm.nih.gov/protein/ (accessed on 20 April 2022). All the collected sequences were used for epitope prediction restricted to HLA-A*02, one of the most prevalent MHC-I haplotypes in the Brazilian population [23], in the NetCTL 1.2 web software available at https://services.healthtech.dtu.dk/service.php?NetCTL-1.2/ (accessed on 20 April 2022). Epitopes of nine amino acids were predicted based on MHC-I affinity, peptide cleavage potential and transport ability by transporter associated with antigen protein complex (TAP) [24]. The combined score of these three factors was used to select immunodominant peptides. Only peptides with combined score values above 0.75 were considered putative epitopes for CD8^+^ T cells and maintained for further analysis.

### 2.2. In Silico Analysis Evaluating Affinity Peptide-HLA-A*02:01, Antigen Processing and Immunogenicity

The evaluation of the combined score for the peptides derived from HTLV-1 Tax sequences was performed followed by an analysis of the independent parameters used for the prediction of epitopes: (i) peptide-HLA-A*02:01 affinity, (ii) peptide cleavage and (iii) peptide transport by TAP protein. Then, the immunogenicity of the peptides was evaluated using the MHC-I Immunogenicity tool available at http://tools.iedb.org/immunogenicity (accessed on 28 April 2022) considering residues 2 and 9 of the peptides as the main anchorage points for binding with HLA-A*02:01 [25].

### 2.3. Assessment of Conserved Residues and Amino Acid Frequencies in HTLV-1 Tax Peptides with Different HLA-A*02:01 Affinity Ranges

For the evaluation of conserved regions in association with the peptide immunodominance, terciles of affinity for peptide-HLA-A*02:01 binding, peptide cleavage and transport by TAP protein were calculated using the values obtained for all Tax peptides. The terciles values allowed the identification of sequences with high, medium and low scores for all three evaluated parameters. The frequency of amino acids in the aligned sequences was calculated using MEGA X: Molecular Evolutionary Genetics Analysis v.10 software [26]. Then, the logo of the sequences was generated in the WebLogo3 web software available at http://weblogo.threeplusone.com (accessed on 17 October 2022) to visualize the most prevalent amino acids in the sequences.

### 2.4. Prediction of Cytokine Production and Analysis of Physicochemical Properties of HLA-A*02:01-Restricted HTLV-1 Tax Peptides

Peptide capability of inducing cytokine production (IFN-γ and IL-4) from Th1 and Th2 axes of immune response was predicted by IFNepitope software available at https://webs.iiitd.edu.in/raghava/ifnepitope/scan.php (accessed on 17 October 2022) and IL4Pred software available at https://webs.iiitd.edu.in/raghava/il4pred/predict.php (accessed on 17 October 2022). In both cases, SVM-based methods were applied. Data regarding physicochemical features, such as hydrophobicity, hydrophilicity, stereochemistry, hydrogen bonds, charge, hydropathicity, isoelectric point (pI), amphipathicity and molecular weight, of peptides were also predicted by IL4Pred software.

### 2.5. Evaluation of the Repertoire of Tax Epitopes with Reactivity to HLA-A*02:01 Using the PEPperCHIP © Microarray System

Evaluation of the HLA-A*02-restricted immunodominant peptide repertoire was performed using PEPperCHIP © custom peptide microarray (PEPperPRINT ©, Heidelberg, Germany). Briefly, the revelation system of the microarray is based on an antibody (mouse IgG) fused to HLA-A*02:01 as a primary reagent that binds to nine amino acid peptides adsorbed in a glass slide. To identify which peptides bind to HLA-A*02:01 and the intensity of binding, we used a secondary reagent, an anti-mouse IgG1 conjugated to the fluorochrome Cy3. The secondary reagent binds to the mouse IgG1 (Fc portion) and is then read out by the microarray reader [27,28]. For the microarray, 345 (9-mer) peptides covering all the extensions of Tax protein were synthesized and adsorbed at 1 nM concentration on spots in duplicates located on a glass slide (75.4 mm × 25.0 mm × 1 mm). The first step of the assay consisted of pre-labeling the microarray glass slide with anti-murine IgG1/Cy3 secondary antibody (Becton Dickinson Biosciences—BD Biosciences, San Jose, CA, USA), for exclusion of background reactivity. For this, the microarray slide was incubated with standard buffer (phosphate buffer saline (PBS) + 0.05% Tween 20; pH 7.4) for 15 min at room temperature. After complete removal of the buffer, a new incubation was performed with blocking buffer (PBS + 0.05% Tween 20 + 1% bovine serum albumin; pH 7.4) for 30 min at room temperature. After complete removal of the buffer, the microarray slide was incubated with anti-murine IgG1/Cy3 secondary antibody (BD Biosciences, San Jose, CA, USA) diluted at a ratio of 1:5000 in staining buffer (standard buffer + 10% blocking buffer) for 45 min at room temperature in the dark. After complete removal of the secondary antibody, the microarray slide was washed three times in standard buffer, dried in airflow and then scanned. For the second stage of the assay, the microarray slide was re-equilibrated by a new incubation with standard buffer for 15 min at room temperature. After complete removal of the buffer, concentrations of 1 µg/mL and 10 µg/mL of HLA-A*02:01:Ig dimeric protein produced in-house [27,28] were diluted in staining buffer and added to the microarray glass slide and incubated for 16 h at 2–8 °C. The microarray slide was washed three times using standard buffer and incubated with anti-murine IgG1/Cy3 secondary antibody (BD Biosciences, San Jose, CA, USA) diluted in a 1:5000 ratio for 45 min at room temperature in the dark. After complete removal of the secondary antibody, the slide was washed three times in standard buffer, dried in airflow and then scanned. Throughout the assay, all incubations were performed under constant agitation in an orbital agitation system (140 rpm). The PEPperCHIP © microarray slide was digitized on the Affymetrix 428 Array Scanner device (Thermo Fisher, Waltham, MA, USA). Data regarding HLA-A2:β2M:Ig reactivity to adsorbed peptides were pre-analyzed using PepSlide^®^ Analyzer (PEPperPRINT ©, Heidelberg, Germany).

### 2.6. Identification of Peptides Position within Full Tax Viral Protein Sequence

To identify the position of the peptides used in the PEPperCHIP © microarray along the HTLV-1 Tax protein, all peptides of each Tax sequence were aligned with the complete sequence of the source virus using the Muscle alignment algorithm [29] in the software Unipro UGENE v.26 [30].

### 2.7. Expression and Purification of the C-Terminal Portion of HTLV-1 Tax

The C-terminal portion of HTLV-1 Tax (C’Tax) is a recombinant protein of approximately 16 kDa resulting from the cloning of the C-terminal portion of the HTLV-1 Tax protein (amino acid sequence 240–350) produced in our laboratory. C’Tax was expressed in pET21-His vector (GenOne) in a prokaryotic model from *Escherichia coli* BL21(DE3). An aliquot of transformed *E. coli* BL21(DE3), stored at −70 °C, was grown in Luria–Bertani (LB) agar (Bacto tryptone 1% *w*/*v*, yeast extract 0.5% *w*/*v*, 1% NaCl 171 mM, 1.5% agar *w*/*v*) with antibiotics (ampicillin 10%). To induce protein production, IPTG (isopropyl-beta-D-thiogalactopyranoside) was added to a final concentration of 1 mM and incubated again in an orbital shaker (New Brunswick Scientific C24—Edison, NJ, USA) for 4 h at 300 rpm, at 37 °C. The samples were centrifuged at 10,000× *g* for 5 min, and the pellet obtained was analyzed by SDS-PAGE. After confirmation of the protein expression, the C’Tax was purified by nickel chelate column chromatography (Qiagen, Hilden, Germany) using ÄKTA Start purification system (GE Healthcare, Chicago, IL, USA) and later the presence of the protein was confirmed by a Western Blot assay. The protein was dialyzed and cleared of LPS and after purification, C’Tax was stored at −20 °C in stock solution of 100 μg/mL.

### 2.8. Obtention of Peripheral Blood Mononuclear Cells

Peripheral blood mononuclear cells (PBMCs) from HLA-A2^+^ HTLV-1 asymptomatic carriers were obtained from Grupo Interdisciplinar de Pesquisas em HTLV (GIPH). The ethics committee has approved the present study. Cells were obtained by a Ficoll-Paque (Sigma-Aldrich, Saint Louis, MO, USA) gradient as recommended by the manufacturer. After thawing, cells were washed with RPMI 1640 medium (Sigma-Aldrich, Saint Louis, MO, USA) and centrifuged for 10 min at 1400 rpm at 18 °C. After centrifugation, cells were counted and adjusted to 1 × 10^7^ cells/mL of RPMI 1640 medium.

### 2.9. C’Tax Cytotoxicity Assessment

For the assessment of cytotoxicity, 500,000 cells were incubated in U-bottomed 96 well plates (Falcon, BD Biosciences, San Jose, CA, USA) for all the assays in the presence and absence of C’Tax for the toxicity experiments and C’Tax culture. After that, cell viability tests were performed with 2 μg/mL of the purified recombinant protein incubated with PBMCs for up to 72 h. For the viability assay, Trypan Blue (0.4%) (Invitrogen, Carlsbad, CA, USA) was used and the results were obtained by flow cytometry. Data analysis was performed using Flowjo analysis software (BD Biosciences, San Jose, CA, USA).

### 2.10. In Vitro Stimulation of Peripheral Blood Lymphocytes with C’Tax

To verify the production of cytokines and CD8^+^ T cell activation of peripheral blood lymphocytes in the face of specific C’Tax stimulation, an aliquot of recombinant protein was thawed on ice at the time of use, diluted in RPMI and incubated with 500,000 cells at 2 μg/mL final concentration for 72 h. As a positive control, PBMCs were cultured with a non-specific stimulation of Concanavalin A (ConA) at a concentration of 10 μg/mL, in the last 4 h of stimulation. Brefeldin was added also for 4 h. Then, 2 mM EDTA was added to the cultures for 15 min, at room temperature and protected from light. After incubation, the plate was centrifuged at 660× *g*, for 10 min, at room temperature, to harvest the supernatant for the determination of cytokines by CBA. Cell pellets were washed for flow cytometric analysis of CD8^+^ T cells.

### 2.11. Flow Cytometry Analysis of CD8^+^ T Cell Activation

After centrifugation, PBMCs were washed with PBS and stained with PE-Cy7 anti-CD3 (clone: SK7), PERCP Cy5.5 anti-CD45RO (clone: HP3G10), AF700 anti-CD8 (clone: RPA-T8) and APC-Cy7 anti-CD69 (clone: FN50) for 30 min at room temperature. After the staining, cells were permeabilized and stained with BV421 anti-human IFN-γ antibody or PE anti-human IL-4 antibody for 30 min at room temperature. After staining, cells were washed with washing solution twice and resuspended in PBS for immediate flow cytometric acquisition. For FACS acquisition, 100,000 lymphocytes were acquired using a BD LSR Fortessa (BD Biosciences, San Jose, CA, USA) flow cytometer. Data acquisition was performed using the FACS DIVA software (BD Biosciences, San Jose, CA, USA) and data analysis using FlowJo software (version 10.6.2, BD Biosciences, San Jose, CA, USA).

### 2.12. Cytometric Bead Array (CBA)

Microspheres coated with monoclonal antibodies specific for target molecules IFN-γ and IL-4 were employed for the quantitative flow cytometric measurement of cytokines in the culture supernatant of C’Tax-stimulated PBMC. This method allows the simultaneous measurement of multiple elements on a single platform. The CBA immunoassay kit (Enhanced CBA kit—Becton Dickinson Biosciences Pharmingen, San Diego, CA, USA) was used for the quantitative analysis of IFN-γ and IL-4, according to the manufacturer’s instructions. Results were expressed in pg/mL defined according to the standard curve for each cytokine.

### 2.13. Statistical Analysis

Statistical analysis was performed using GraphPad Prism v.8.0 software (GraphPad Software, San Diego, CA, USA). Non-parametric data distribution was confirmed by the Shapiro–Wilk test. Wilcoxon test was applied for statistical comparisons involving only two groups. Multiple comparisons were performed using the Kruskal–Wallis method followed by Dunn’s post-test, in the case of unpaired data, and the Friedman method with Dunn’s post-test to compare paired data. Chi-square test was applied for the comparison of unpaired data regarding the number of reactive peptides. Spearman’s rank correlation test was applied for the correlation of in vitro reactivity and in silico prediction of cytokine induction. ROC curve analysis and performance indices were evaluated using GraphPad Prism v.8.0 software (GraphPad Software, CA, USA). Differences were considered statistically significant at *p* < 0.05.

## 3. Results

### 3.1. Diversity of HLA-A*02:01 Reactivity for Immunodominant Peptides from N-Terminal, Central and C-Terminal Regions of Tax

Figure 1 demonstrates the results of HLA-A*02:01 reactivity for immunodominant peptides from the N-terminal (blue), Central (red) and C-terminal (green) regions of Tax. The results show clearly high HLA-A*02:01 reactivity of peptides in all three regions along the Tax protein, which clearly indicates abundant immunodominant sequences. Sequential hotspots for HLA-A*02:01 reactivity are observed in the initial region of N-terminal Tax (Figure 1A). An intercalate pattern of high HLA-A*02:01 reactivity is observed for the Central and C-terminal regions (Figure 1B,C).

### 3.2. Dose-Dependent HLA-A*02:01 Reactivity for Immunodominant Peptides from N-Terminal, Central and C-Terminal Regions of Tax

In order to verify the ability of Tax peptides to bind in a dose-dependent manner, the microarray (PEPperPRINT ©, Heidelberg, Germany) was designed and generated. The peptide array employed a single concentration of peptides at 1 nM and three different points (0, 1 μg/mL, 10 μg/mL) of a dimeric protein containing HLA-A*02:01 domain fused to β2-microglobulin (HLA-A2:β2M:Ig Protein) produced in-house. As shown in Figure 2, global analysis of the intensity of fluorescence for peptide-HLA-A*02:01 binding indicated a dose–response binding at a single concentration of peptide in the array (Figure 2A). The same pattern was observed for peptides from all three regions of Tax protein.

However, the HLA-A*02:01 reactivity of peptides from the N-terminal region was superior to the Tax peptides from the C-terminal region at 1 μg/mL of HLA-A2:β2M:Ig Protein (Figure 2B). This difference was not observed at 10 μg/mL of HLA-A2:β2M:Ig Protein. No difference was observed with the Central region. Although lower median fluorescence intensity was observed in C-terminal peptides, there was no significant difference in reactivity at 10 μg/mL of HLA-A2:β2M:Ig Protein, which may indicate that higher availability of HLA-A*02:01 decreased the differential reactivity pattern observed by C-terminal Tax peptides among peptides from other regions (Figure 2C).

### 3.3. Reactivity of Tax Peptides from N-Terminal, Central and C-Terminal Regions of Tax According to Immunodominant Tax^11–19^ Peptide

Figure 3 shows the results for the analysis of the HLA-A*02:01 reactivity of Tax peptides from N-terminal, Central and C-terminal regions, taking the immunodominant Tax^11–19^ peptide as a gold standard for HLA-A2 binding. For that, the fold-change of HLA-A*02:01 reactivity (at 1 and 10 μg/mL) was calculated for each peptide from the N-terminal, Central and C-terminal regions of Tax over the HLA-A*02:01 reactivity of Tax^11–19^ (Peptide reactivity/Tax^11–19^ reactivity) (Figure 3A).

The results of fold-change reactivity obtained at lower concentrations of HLA-A*02:01 reveal that peptides from the three regions contain sequences that react significantly more than Tax^11–19^. At higher concentrations of HLA-A*02:01, the fold-change reactivities of N-terminal, Central and C-terminal regions are significantly decreased when compared to the ones at lower concentration of HLA-A*02:01 (Figure 3A). In addition, we explored the absolute number of reactivity of peptides whose performance in both HLA-A*02 concentrations surpassed the reactivity of the immunodominant peptide Tax^11–19^. High binding performance was observed at low HLA-A*02 concentration for the binding of peptides located along the N-terminal (*n* = 41), Central (*n* = 49) and C-terminal (*n* = 52) portions (Figure 3B). Subsequently, a sequence of the top 10 reactive peptides identified by the first three amino acids was selected and is shown in Figure 3C. These peptides exhibited significantly higher performance regardless of HLA-A*02 concentration as compared to the reactivity for Tax^11–19^ peptide. The results show that all top 10 peptides react over 1000 times more than Tax^11–19^ peptide at low HLA-A*02 concentration and around 1 fold-change at high HLA-A*02 concentration. The in silico analyses for HLA-A*02 and HLA-A*24 haplotypes, as well as predictive parameters of MHC class I affinity, TAP protein binding, cleavage and combined score (a combination of the previous three parameters), show that the top seven peptides demonstrate positive scores for HLA-A*02, being peptide #7 (GSV) the one with the best performance (Figure 3C). For HLA-A*24 prediction scores, the top four peptides show positive prediction scores for all parameters, being peptide #3 (DWC) the one with the best prediction scores.

These results show plural Tax peptides that exert potent binding with superior reactivity to the Tax^11–19^ at physiological baseline expression of HLA-A*02:01. However, during instances in which human MHC expression may be augmented such as inflammatory conditions, Tax^11–19^ may remain as the hallmark of immunodominance mediated by Tax protein.

### 3.4. Amino Acid Profile of Reactive HLA-A*02:01-Restricted Epitopes Suggests Different Patterns in N-Terminal, Central and C-Terminal Tax Peptides

In order to understand the different patterns observed for the N-terminal, Central and C-terminal Tax peptides, we sought to evaluate the profile of amino acids that comprise the immunodominant Tax peptide sequences. Figure 4 shows the results for the frequency of amino acids of different biochemical properties (non-polar, polar, acid and basic) of HLA-A*02:01-reactive peptides. Overall, the assessment of amino acid patterns within peptides that reacted to HLA-A*02:01 in the peptide microarray revealed that the three regions of Tax present peptide sequences with distinct classes of amino acids.

N-terminal peptides display heterogeneous distribution of non-polar amino acids along the sequence, with these amino acids concentrated at the second and ninth positions [31], contrasting with Central and C-terminal regions. C-terminal peptides show a higher frequency of polar and charged amino acids as compared to the Central and N-terminal regions (Figure 4A). Sequence logo analysis allowed the identification of predominant amino acid residues in each region of Tax (Figure 4B). The N-terminal region has peptides with abundant glycine (gly), leucine (leu) and proline (pro). Central region peptides display higher frequencies of leu and pro, while C-terminal peptides display higher frequencies of leu, pro and glutamic acid (glu) residues (Figure 4B).

### 3.5. Immunomodulatory Properties of the HLA-A*02:01-Reactive Tax Peptides by In Silico Analysis of the IFN-γ Axis vs. IL-4

Tax is an HTLV-1 protein capable of inducing a robust response associated with immune cell activation featured by the production of proinflammatory and immunomodulatory cytokines. Thus, an in silico evaluation of potential N-terminal, Central and C-terminal Tax peptides in inducing a response mediated by interferon-gamma (IFN-γ) and interleukin-4 (IL-4) was carried out. For this analysis, the results of the in silico analysis of IFN-γ (Appendix A) and IL-4 (Appendix A) indicated that HLA-A*02:01-reactive Tax immunodominant peptides can induce an efficient and robust production of these cytokines. N-terminal sequences have higher frequencies of IFN-γ inducers, while C-terminal sequences have higher frequencies of IL-4 inducers (Figure 5A).

The Central region presents the higher frequencies of peptides that are non-inducers for both cytokines. These results are not dependent on the HLA-A*02:01 concentration (Figure 5A). However, after removing non-inducers from the dataset, the Central region presents the highest scores of IFN-γ production, while the C-terminal region presents the highest score of IL-4 production (Figure 5B). The ROC curve analysis showed high performance and significance (*p* < 0.05) in distinguishing cytokine-inducing and non-inducing peptides with high sensitivity, specificity and elevated accuracy (AUC 0.89, Figure 5C).

### 3.6. Integrative Analysis of Physicochemical and Immunogenic Features of HLA-A*02:01-Reactive Immunodominant Peptides from the N-Terminal, Central and C-Terminal Regions of Tax

The results in Figure 6 show the integrative analysis based on heatmaps displaying the correlation matrix of physicochemical and immunological properties of HLA-A*02:01-reactive Tax immunodominant peptides. Data regarding the integration of physicochemical features included hydrophobicity, hydrophilicity, steric hindrance, net hydrogen, side bulk, charge, hydropathicity, isoelectric point (pI) and molecular weight of Tax peptides. HLA-A*02:01 reactivity and cytokine prediction scores were also included in the heatmap analysis.

The results for the full Tax protein demonstrated that there was no direct correlation between the results of IFN-γ and IL-4 scores for the HTLV-1 Tax immunodominant peptides of the N-terminal and Central regions (Figure 6B,C). The results for N-terminal peptides demonstrated that the HLA-A*02:01 reactivity correlated directly with the IFN-γ score as well as with the hydrophobicity score and related physicochemical properties (Figure 6B). On the other hand, HLA-A*02:01 reactivity and IFN-γ correlate inversely with the IL-4 production score (Figure 6C). Conversely, a direct correlation between HLA-A*02:01 reactivity results and the IL-4 production score is observed for peptides in the C-terminal portion of Tax, which demonstrates a highly peptide-specific correlation of physicochemical and immunogenic features of Tax peptides (Figure 6D).

### 3.7. Activation of CD8^+^ T Cells Upon Stimulation with Recombinant C-Terminal Portion of Tax—Induction of a Mixed Pro-Inflammatory and Modulatory Profile

Considering the results obtained after the in silico analysis, we focused on understanding the effect of the C-terminal portion of the HTLV-1 Tax protein in CD8^+^ T cell activation. For that, we produced a recombinant protein named C’Tax containing the amino acid region from 240 to 350 of Tax protein in a heterologous system. Low cytotoxicity of this protein was observed in human peripheral blood mononuclear cells with 99% viability at up to 2 μg/mL (Appendix A). Figure 7 shows the results of CD8^+^ T cells upon stimulation with C’Tax.

Upon stimulation with C’Tax of peripheral blood mononuclear cells from HLA-A2^+^ HTLV-1 carriers, robust activation of CD8^+^ T cells (Figure 7A) was observed, including the memory T cell compartment that includes antigen-experienced T cells characterized by CD45RO^+^CD8^+^ T cells (Figure 7B). Furthermore, increased frequency of both IFN-γ^+^ and IL-4^+^ CD8^+^ T cells was observed upon stimulation with C’Tax (Figure 7C). Supernatants of PBMCs from HTLV-1 carriers cultured with C’Tax also presented elevated levels of soluble secreted IFN-γ and IL-4 (Figure 7D), corroborating the results of intracellular cytokine staining. These results indicated that this recombinant protein is able to activate CD8^+^ T cells and induce two distinct profiles in those cells, as predicted in the in silico analysis.

## 4. Discussion

Despite several attempts of development of immunogens and therapeutic approaches, vaccines for ATL and HAM/TSP are still far from being achieved. The treatment of HAM/TSP is basically symptomatic, with the use of corticosteroids for more severe cases. For ATL, conventional chemotherapy has short-term efficacy and may lead to clonal selection of neoplastic cells transformed by the viral genome that are more aggressive in their proliferation. Allogeneic stem cell transplantation offers long-term disease control for only a portion of transplanted patients, but a few are able to reach transplantation due to viable donor compatibility and intrinsic status for the transplant. This has led, in recent years, to the performance of a series of clinical trials with new treatments, which until now have shown results below what is necessary for ameliorating HTLV-1 inflammatory diseases [32].

In the present study, a fluorescence HLA-A*02:01-based Tax peptide microarray was designed and developed for the mapping of HLA-A*02:01-restricted immunodominant epitopes within the N-terminal, Central and C-terminal regions of the polyprotein. The results identified several immunodominant HLA-A*02:01-reactive peptides, which are either IFN-γ or IL-4 inducers. In humans, the balance of proinflammatory and regulatory responses is strongly associated with a protective profile that may prevent or delay the development of HAM and ATL. In this context, it is necessary to understand the biological properties of Tax protein, in search of new targets and for the development of therapeutic vaccines. More studies are still needed to understand the Tax-mediated mechanisms of transformation and infection of HTLV-1, as well as the processes of immune recognition of this protein.

Tax is the most studied viral transcriptional activator in HTLV infection, and its expression occurs early in the viral infection [33,34,35]. This viral protein activates transcription of the viral genome and several cellular pathways involved in cell proliferation. Previous reports established Tax as a viral oncogenic transforming cell agent through a variety of mechanisms, including chromosomal instability [34,36], downregulation of DNA repair mechanisms [34,37,38], activation of dependent kinases [34,39], dysregulation of NF-κΒ [34,40,41] and Akt signaling pathways [39,42] and p53-tumor-suppressor protein silencing [34,43,44]. In addition to the above-mentioned effects, the HTLV-1 Tax protein may play a role in pro-apoptotic mechanisms, because Tax acts to control the expression of regulatory cell survival genes [36]. Importantly, Tax acts as the primary viral antigen that is recognized by the host’s immune system, especially in the context of the HTLV-1-specific anti-viral CD8^+^ T-cell responses. During chronic infection, genetic and epigenetic inactivation of the Tax gene through mutations, promoter hypermethylation and promoter deletion is observed as the disease progresses to promote continued viral persistence [34,45,46].

In fact, addressing viral reservoirs and viral persistence requires intimate knowledge of anti-viral mechanisms within the infected host, such as virus-primed CD8^+^ T cells, that represents the most varied oligoclonal composition of blood lymphocytes of ATL and HAM/TSP patients [22]. The expansion of oligoclonal T cell phenotype is highly diverse in CD8^+^ T cells, diverging from the CD4^+^ phenotype, which may indicate that CD8^+^ T cells may play a role in combating the virus [22,47]. Therefore, putative control of the HTLV-1 proviral load is associated with robust cytotoxic CD8^+^ lymphocyte activity in the peripheral blood of humans [21] and non-human primate animal models [47]. In this sense, harnessing CD8^+^ T-cell responses in situ and in the periphery is an attractive target for the development of therapeutic vaccines in a haplotype-specific manner.

Regarding haplotypes, there is an association between HLA haplotype and HTLV-1-related diseases. Previous studies demonstrate that HLA-A*02 expression is associated with diminished proviral load and reduced risk of developing HAM/TSP in HTLV-1 carriers. Interestingly, in this same study, it was observed that the class II HLA-DR*0101 allele offered greater protection together with the HLA-A*02 allele. On the other hand, another study demonstrated that HLA-Cw*08 was also associated with disease protection, and the HLA-B*5401 haplotype was associated with susceptibility to HAM/TSP [48,49]. Fabreti-Oliveira et al. [23] carried out an extensive assessment of the composition of HLA haplotypes present in the Brazilian population, identifying HLA-A*02:01 as the one with the highest incidence in the national territory. These results are extremely important as they indicate that national studies evaluating the CD8^+^ T response should consider restrictions associated with this specific haplotype to ensure a representative view of the immunological status of the population in HTLV-1 endemic areas. To assess the prevalence of HLA-A*02-restricted viral peptides, peptide prediction was performed for the most observed HLA haplotype in Brazil and Japan, that is, HLA-A*02 (0.2332) in addition to HLA-A24 and HLA-DR (MHC-II) [23,50,51]. The peptides generated in this study performed well in the in silico analysis for binding to HLA-A*02 and HLA-A24, but not well to HLA-DR. It is possible that the nine amino acid sequences were too small to bind to the HLA-DR groove that can accommodate larger peptides in its MHC class II groove.

The interaction between the peptide/MHC-I complex and the TCR is the first step in the activation of CD8^+^ T cells. However, different MHC-I haplotypes are likely to have distinct combinations of peptides with different anchor residues, based on the sequence present in the binding cleft. Such diversity affects the quaternary structure of the peptide/MHC-I complex, leading to variability in the recognition of peptides by the TCR and affecting the cellular immune response [29,52,53,54,55].

Tax-specific T lymphocytes are abundant in HTLV-1-infected individuals and have been suggested to contribute to the control of infected cells. Within this context, the N-terminal portion of the Tax protein known as Tax^11–19^ was reported as the most immunogenic epitope of Tax, capable of inducing a robust response of specific Tax^11–19^-specific CD8^+^ T cells [17,29,56]. Despite this evidence, it remains unknown whether anti-Tax cytotoxic T lymphocytes control the proliferation of infected cells in vivo, but recent clinical studies have shown the efficacy of Tax-targeted vaccines in the maintenance of long remission of ATL patients [57]. In addition to that, other studies have found Tax epitopes, other than Tax^11–19^, able to bind to HLA-A2 and induce CTL responses [58,59,60,61]. Therefore, we evaluated the results of HLA-A*02:01 reactivities obtained for all peptides from N-terminal, Central and C-terminal regions and compared to the scores of HLA-A*02:01 reactivity for Tax^11–19^ (LLFGYPVYV). The results demonstrate that in higher HLA-A*02:01 expression, Tax^11–19^ showed elevated reactivity superior to several other peptides. Therefore, during instances in which human MHC expression may be augmented such as inflammatory conditions, Tax^11–19^ may remain as the hallmark of immunodominance mediated by Tax protein. However, the results also indicate that other Tax peptides from N-terminal, Central and C-terminal regions show strong HLA-A*02:01 binding with 1000 times higher fluorescence to the Tax^11–19^ reactivity at physiological baseline expression of HLA-A*02:01. These results may indicate the importance of alternative Tax peptide search for the rational design of prophylactic and therapeutic vaccines. In agreement with our findings, previous immunoproteomics studies have identified novel epitopes directly presented by HTLV-1-infected CD4^+^ T cells expressing immunodominant HLA-A2 and HLA-A24 haplotypes. In this study, the novel peptides outperformed Tax^11–19^. In fact, Tax^11–19^ epitope was not identified as the high-affinity binder but rather exhibited low binding [59], possibly because of the lower physiological HLA expression of the immunoproteomics-based assays.

Previous epitope mapping studies support the search for new Tax immunodominant epitopes in addition to canonic Tax^11–19^ peptide [17,29,58,59,60,61]. It is important to mention that previous studies have focused on Tax epitopes that have strong association with high INF-γ production that in turn associates with high proviral load in HAM/TSP patients [60,61]. Therefore, these peptides are probably linked to exacerbated inflammation rather than controlling viral reservoirs.

The results of the biochemical characteristics of the Tax epitopes showed that the most frequent amino acid is the non-polar amino acid, leucine, in the polypeptide sequence. In a similar way, the 10 most frequent amino acids observed were evaluated for their distribution in the polypeptide chain, being predominant in those of non-polar or polar uncharged biochemical characteristics. Leucine distribution results reveal that there is a cumulative frequency at positions 2–3 and 8–9 of the polypeptide chain. The results for the other nine most frequent amino acids indicate that their distribution is non-contiguous, that is, non-sequential along the peptide. The importance of residues 2 and 9 for the anchoring of peptides to HLA-A*02 is already described and known [29]. Our results using new predictor tools indicated that conserved leucine (L) residues at positions 2 and 9 are important for increasing peptide/HLA-A*02 affinity.

The general prevalence of non-polar amino acids in the aligned peptides agrees with several previous studies, which associate hydrophobic peptides not only with peptide/MHC-I binding but also with the induction of the CD8^+^ T response [52,53,54,55].

The late and most recent in vivo evidence of the putative role of Tax peptides indicates high pleiomorphic activity of Tax, with the elevated potential of inducing protective immunity. In order to expand the knowledge about the immunomodulatory potential of the Tax protein, the prediction of the potential for the production of pro-inflammatory and regulatory cytokines was made by in silico analysis of IFN-γ and IL-4, which indicated the existence of peptides capable to induce the production of both cytokines efficiently and robustly. There is an inverse trend in the production of these cytokines, indicating that the same peptide could have the property of inducing the production of both IFN-γ and IL-4 by T cells. In this sense, it was observed that the C-terminal region presents the majority of the immunodominant IL-4 inducers. These results were corroborated by in vitro studies showing CD8^+^ T cell activation upon stimulation with C’Tax. In addition, C’Tax stimulus induces a mixed profile composed of both proinflammatory and regulatory cytokines, which confirms the plural features of CD8^+^ T cell-mediated immune response by C-terminal Tax peptides, which could ultimately modulate HTLV-infected lymphocytes and prevent activation and migration of those cells to the central nervous system and important vital sites.

In fact, specific C-terminal Tax epitopes have been reported as highly immunogenic to HLA-A24:02-restricted T cells with a focus on Tax^301–309^ (SFHSSLHLLF), which shows strong cytolytic response [61]. In addition, C-terminal Tax epitopes are also highly recognized by HTLV patient serum antibodies with a focus on peptides Tax^316–335^, Tax^331–350^ and Tax^336–353^ [62]. Hotspots were found within 240–267, 291–303 and 327–354 regions, which contain the top C-terminal peptides. Highlight was observed in C-terminal peptide TLTTPGLIW (Tax^240–249^) which exhibited a superior profile in HLA binding as compared to Tax^11–19^. Therefore, in addition to the role of the C-terminal region of Tax in activating CD8^+^ T cells, this region could also contribute to inducing both arms of immune response in a future vaccine formulation. In fact, proper vaccination protocols should aim at not only controlling viral spread but also modulating inflammation within the host during chronic infection, which ultimately will decrease the chances of exacerbated inflammation and tissue damage mediated by Tax at the late stages of infection.

In summary, these results support and shed light on the search for new Tax immunodominant epitopes with both proinflammatory and regulatory dominance, in addition to the canonic inflammatory Tax^11–19^, for the rational design of immunomodulatory vaccine strategies for HTLV-1 chronic diseases.

## Figures and Tables

**Figure 1 biomolecules-13-00545-f001:**
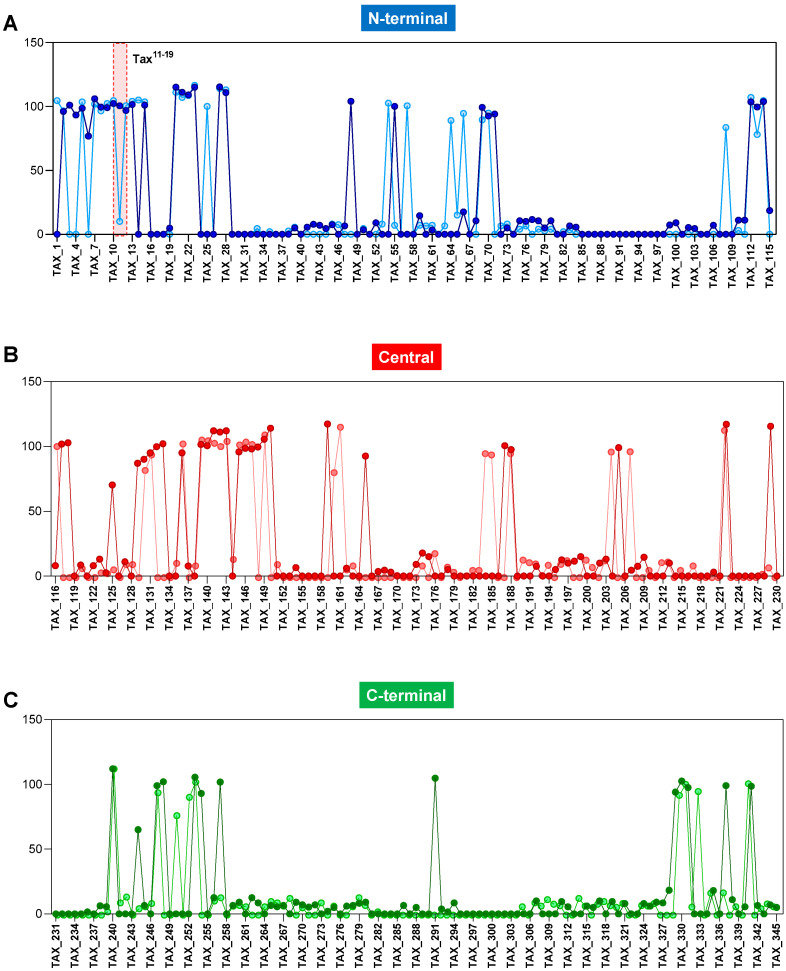
Reactivity of peptides restricted to the HLA-A*02:01 haplotype distributed along the N-terminal (**A**), Central (**B**) and C-terminal (**C**) portions of Tax. Light colors represent reactive peptides with 1 µg/mL of HLA-A*02:01:Ig dimeric protein, and dark colors represent reactive peptides with 10 µg/mL of HLA-A*02:01:Ig dimeric protein. The red rectangle in the dotted line, present in (**A**), represents the HLA-A*02:01 reactivity of the Tax^11–19^ peptide present in the N-terminal region of Tax.

**Figure 2 biomolecules-13-00545-f002:**
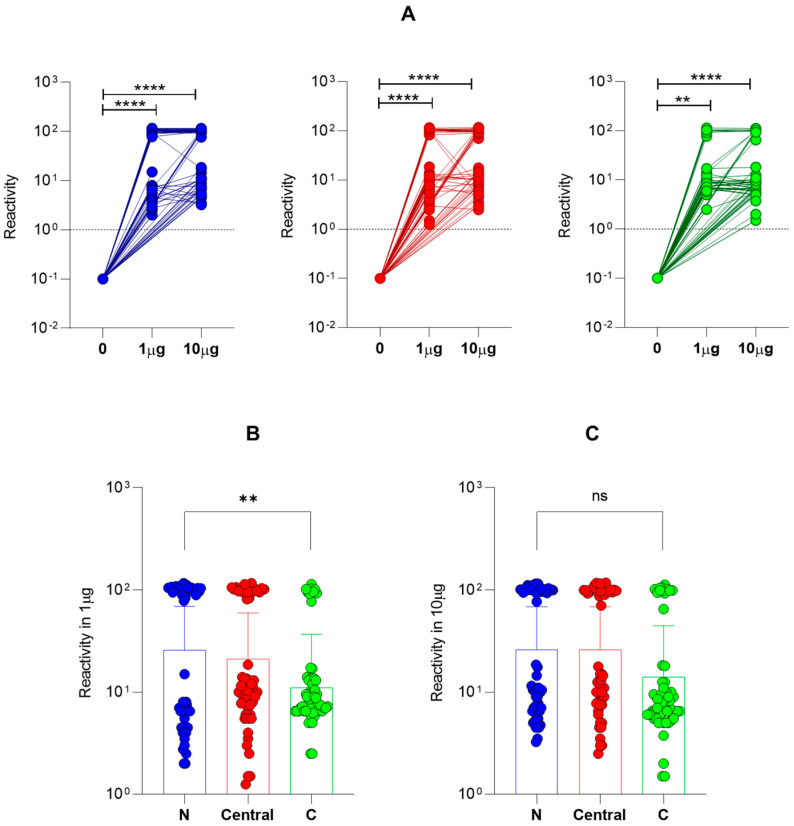
Dose-dependent binding of Tax peptides. Dose-dependence evaluation of HLA-A*02:01 haplotype-reactive Tax peptides distributed in the N-terminal, Central and C-terminal regions of the Tax protein. (**A**) Assessment of the reactivity of Tax peptides to HLA-A*02:01 using 0, 1 and 10 μg/mL distributed in the N-terminal, Central and C-terminal regions. (**B**) Scatter-plot representation of peptides reactive to HLA-A*02:01 at a concentration of 1 μg/mL distributed among the three regions of Tax: N-terminal, Central and C-terminal. (**C**) Scatter-plot representation of peptides reactive to HLA-A*02:01 at a concentration of 10 μg/mL distributed among the three regions of Tax: N-terminal, Central and C-terminal. Two asterisks (**) indicate *p* ≤ 0.01 and four asterisks (****) indicate *p* ≤ 0.0001.

**Figure 3 biomolecules-13-00545-f003:**
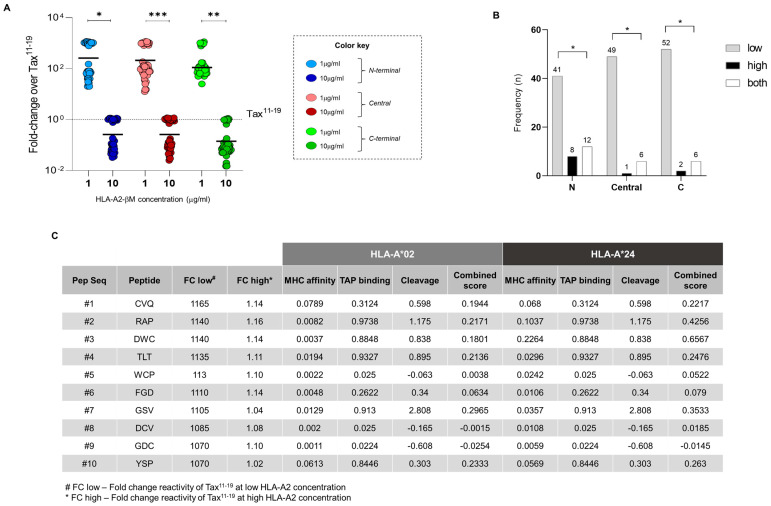
Reactivity of Tax peptides according to immunodominant Tax^11–19^ peptide. Reactivity of HLA-A*02:01 haplotype-restricted peptides compared with Tax^11–19^ reactivity. (**A**) Scatter-plot representation of HLA-A*02:01-restricted peptides at a concentration of 1 and 10 μg/mL distributed among the three regions of Tax: N-terminal, Central and C-terminal. (**B**) Number of peptides located in the N-terminal, Central and C-terminal regions of Tax that showed greater reactivity than Tax^11–19^, considering low, high and both HLA-A*02 concentrations. Connecting lines with asterisks show statistical differences at *p* < 0.05 (* indicates *p* ≤ 0.05; ** indicates *p* ≤ 0.01; *** indicates *p* ≤ 0.001), using Chi-square statistical test. (**C**) Representative table of the top 10 Tax peptides whose reactivity was greater than Tax^11–19^ with in silico prediction data for HLA-A*02 and HLA-A*24 haplotypes.

**Figure 4 biomolecules-13-00545-f004:**
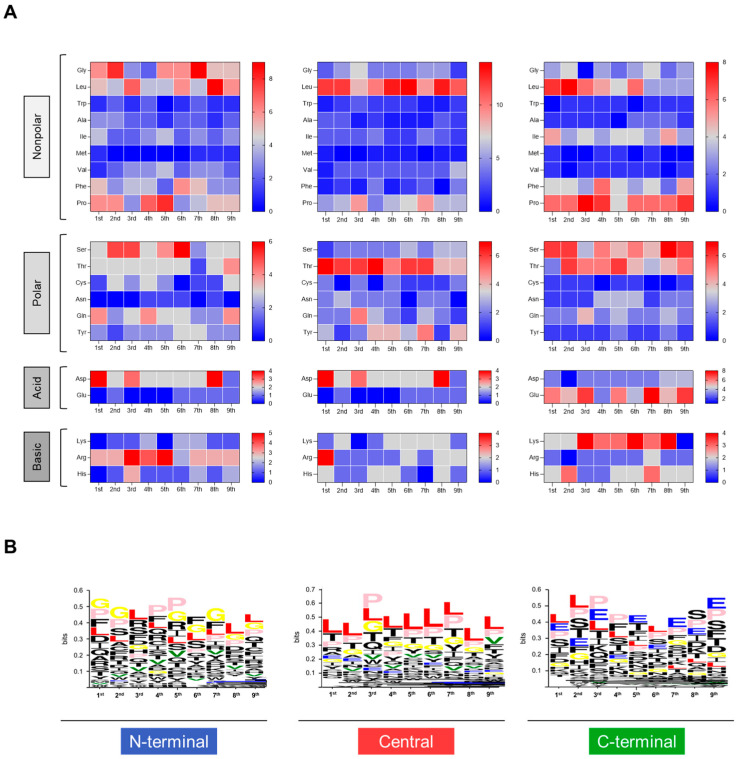
Distribution of different classes of amino acids that constitute reactive peptides according to different Tax regions. (**A**) Frequency of different classes of amino acids represented by heatmap according to the N-terminal, Central and C-terminal portions of Tax. (**B**) Frequency of amino acids from aligned peptide sequences generated in WEBLOGO according to the N-terminal, Central and C-terminal portions of Tax. Leucine (L): red; Valine (V): green; Glycine (G): yellow; Glutamic acid (E): blue; Proline (P): light pink.

**Figure 5 biomolecules-13-00545-f005:**
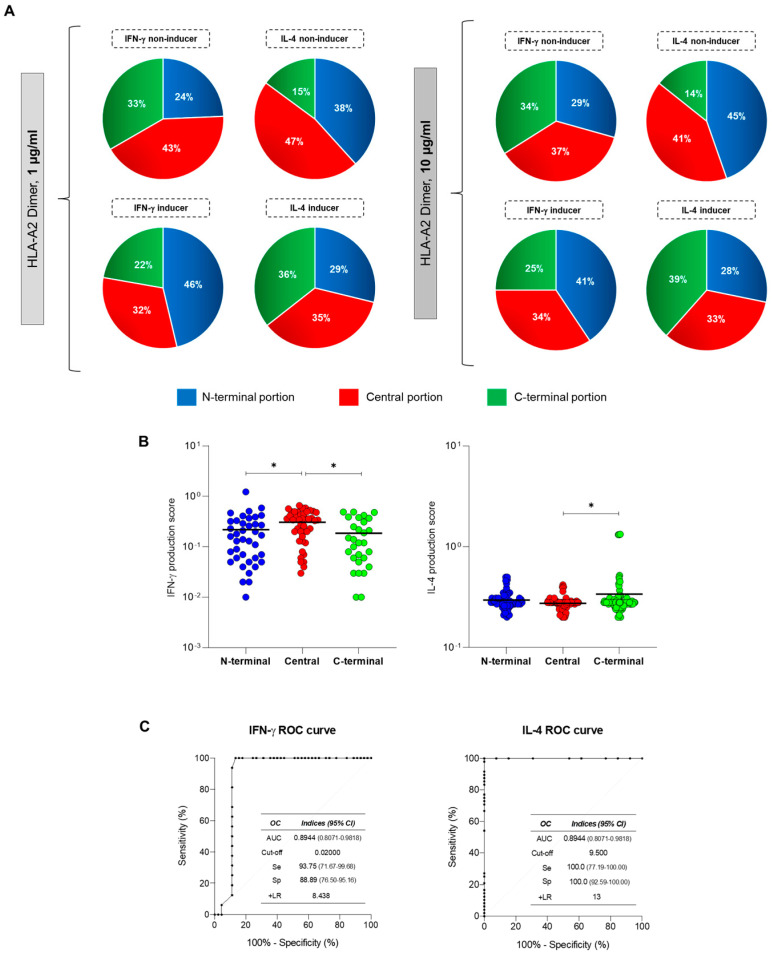
In silico analysis of the functional features of reactive HLA-A*02:01-restricted Tax peptides. Proinflammatory (IFN-γ) and regulatory (IL-4) cytokine profile of HLA-A*02:01-reactive peptides distributed along the N-terminal, Central and C-terminal regions of Tax. (**A**) Percentage of IFN-γ and IL-4 inducing and non-inducing peptides using 1 and 10 μg/mL of the HLA-A*02:01 haplotype. (**B**) Pro-inflammatory and regulatory cytokine production scores. (**C**) Analysis of sensitivity and specificity parameters using ROC curve analysis based on IFN-γ and IL-4 production scores. Connecting lines with asterisks show statistical differences at *p* < 0.05 (* indicates *p* ≤ 0.05).

**Figure 6 biomolecules-13-00545-f006:**
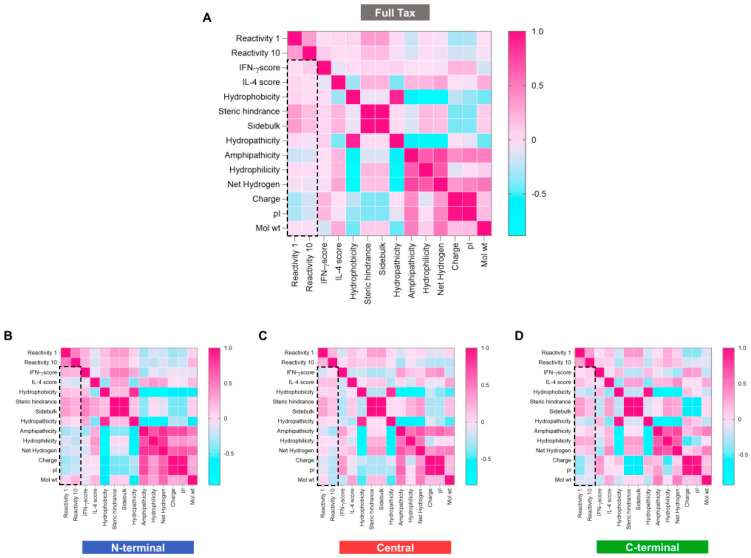
Correlations among physicochemical properties of HLA-A*02-restricted peptides versus reactivity of peptides to the HLA-A*02-01 haplotype at low (1 μg/mL) and high (10 μg/mL) concentrations. Spearman correlation demonstrated by heatmap according to full Tax protein (**A**) and N-terminal (**B**), Central (**C**) and C-terminal (**D**) regions. The black rectangle in the dotted line represents the HLA-A*02:01 reactivity of Tax peptides.

**Figure 7 biomolecules-13-00545-f007:**
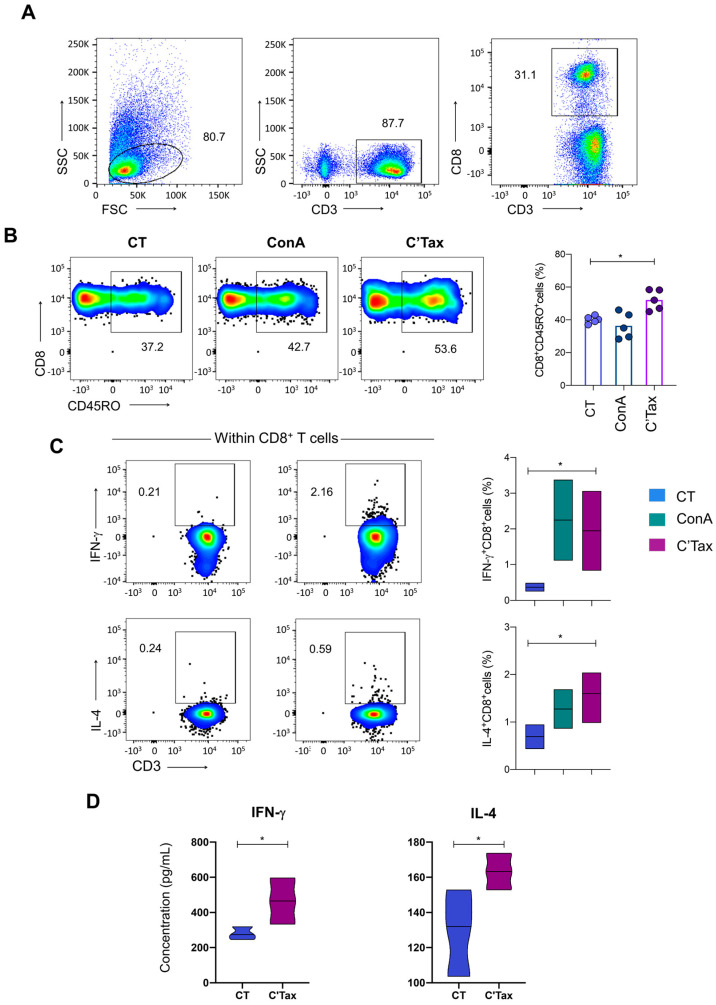
Activation of CD8^+^ T cells upon stimulation with recombinant C-terminal portion of Tax. (**A**) First, the selection of CD8^+^ T lymphocytes (CD3^+^CD8^+^) within total lymphocytes was performed. (**B**) Second, the evaluation of antigen-experienced memory CD8^+^ T cells (CD3^+^CD8^+^CD45RO^+^) after stimulation with C’Tax was carried out and the results are shown as pseudo color plots (left panels) and bar graphs overlaid by scatter plots (right panels). (**C**) After that, IFN-γ and IL-4 production by CD8^+^ T cells was examined and the results are expressed in the flow charts (left panels) and by floating bars (min–median–max values on the right panel). (**D**) Finally, the levels of IFN-γ and IL-4 in PBMC culture supernatants after stimulation with C’Tax were evaluated. The results are plotted as violin graphs. Statistical differences for all plots were indicated by connecting lines and asterisks when *p* < 0.05 (* indicates *p* ≤ 0.05).

## Data Availability

Peptide Microarray Dataset will be available upon reasonable request.

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
