# Peer review of "Diversity of HLA-A2-Restricted and Immunodominant Epitope Repertoire of Human T-Lymphotropic Virus Type 1 (HTLV-1) Tax Protein: Novel Insights among N-Terminal, Central and C-Terminal Regions"

_biomolecules, 2023, doi:10.3390/biom13030545_

Round 1

Reviewer 1 Report

<General Comments>

This is a study in an important direction that reexamines the Tax peptide that activates CTLs for the entire region of Tax. However, compared to previous reports indicating that Tax11-19 is important for CTL activation, the progress of the current study is not clear. It is unclear at what level of HTLV-1 infection a new Tax epitope can be utilized for vaccination in HTLV-1 infected individuals. If authors aim to develop Tax vaccines specific for inhibiting the early infection or chronic infection, they should present evidence of a correlation between Tax-CTL reactivity to N-, Central, or C-Tax epitope in HTLV-1 infected individuals. For example, is Tax-CTL reactivated against N-Tax mainly in early infection and against C-Tax in chronic infection?

At this stage, it seems somewhat premature to "suppress diseases caused by chronic infection" by newly identified Tax epitopes as vaccination. It takes 50 years to develop HAM/TSP and 65+ years to develop ATL. The changes within HTLV-1-infected cells leading up to disease onset are complex and have not been clarified completely. There are also many aspects of the host immune response during this period that are not understood. The experimental design and interpretation of the data should take into consideration whether the goal is to suppress the infection itself or the step where infected cells become inflammation-inducible or tumorigenic decades after infection. In this context, the results of this study will be more convincing if the relationship between infected cells and immune responses in an animal model is presented as a set. Otherwise, the authors should discuss more about the mode of Tax-CTL activation by different Tax epitopes and realistic infected cells in carriers/patients.

<Specific Comments>

1. Line 102: The page is not found.

2. Line 147; Why was anti-mouse IgG/Cy3 secondary antibody used for human HLA-A02:01 microarray?

3. Line 375-379: Although Tax plays a critical role in viral replication in the early stage of infection, its role in the maintenance of chronic infection is limited. At which stage the Tax vaccination would be applied?

4. Line 390: Tax also induces apoptosis.

5. Line 439-447: Any relationship between HLA haplotype and HTLV-1 related diseases?

6. Line 481-483: The interesting result that the C-terminal side of Tax is important as an IL-4 inducer from CTLs should be discussed in terms of the relationship between infected cells and the immune system during the early and chronic infection period of HTLV-1.

7. Line 409-419: These are the main findings of the current study. Most of part of the discussion is a good review of Tax CTL, but more discussion to evaluate the current study is necessary.

Author Response

Dear Reviewer #1,

Please, find below the responses to your queries. We have modified the text in the revised version of the manuscript and modifications were marked in yellow.

REVIEWER#1

Comments and Suggestions for Authors

<General Comments>

This is a study in an important direction that reexamines the Tax peptide that activates CTLs for the entire region of Tax. However, compared to previous reports indicating that Tax11-19 is important for CTL activation, the progress of the current study is not clear.

Response: We would like to thank the Reviewer for this comment. We have worked out to improve the quality of the manuscript and to make the progress of the current study clearer.

It is unclear at what level of HTLV-1 infection a new Tax epitope can be utilized for vaccination in HTLV-1-infected individuals. If authors aim to develop Tax vaccines specific for inhibiting the early infection or chronic infection, they should present evidence of a correlation between Tax-CTL reactivity to N-, Central or C-Tax epitope in HTLV-1 infected individuals. For example, is Tax-CTL reactivated against N-Tax mainly in early infection and against C-Tax in chronic infection?

Response: We appreciate the Reviewer’s comment, which is very important in the discussion regarding early and late chronic HTLV-1 infection. In fact, it is our main point in the manuscript to advise Tax epitopes, other than Tax11-19, as antigenic components for the rational formulation of future therapeutic vaccines to treat HTLV-1 infection and prevent worsening of neurological tissue damage, especially in HAM/TSP. It would be very interesting to know if Tax-CTL is in fact reactivated against N-Tax in early infection and against C-Tax in chronic infection as proposed by the Reviewer, however, early HTLV-1 infection is very difficult to find and track amongst HTLV-1-infected patients even in endemic areas and diagnosis usually starts at very late stages of infection. Therefore, studying CD8+ T cell responses in early infection becomes impractical and, therefore, impossible to be assayed. In addition, the applicability of a vaccine for controlling viral spread during early infection would be also impractical since patients at this stage are rarely diagnosed. We agree with the Reviewer that C-terminal-Tax could be a very attractive target for future immunotherapeutics of chronic HTLV-1 infection. Therefore, we have included an additional figure (new figure 7) to address this point (pages 13 and 14). Reactivation studies need more time and detailed assays to be concluded and should include a larger cohort of patients and/or in vivo studies. Therefore, we hope to address this important topic in a future publication. We believe that future formulations with C’Tax could contribute as vaccine candidates for the therapeutic intervention to treat chronic HTLV-1-infected carriers. 

At this stage, it seems somewhat premature to "suppress diseases caused by chronic infection" by newly identified Tax epitopes as vaccination. It takes 50 years to develop HAM/TSP and 65+ years to develop ATL. The changes within HTLV-1-infected cells leading up to disease onset are complex and have not been clarified completely. There are also many aspects of the host immune response during this period that are not understood. The experimental design and interpretation of the data should take into consideration whether the goal is to suppress the infection itself or the step where infected cells become inflammation-inducible or tumorigenic decades after infection.

Response: We agree with the Reviewer’s comment. We believe that total suppression of infection itself is not possible yet considering possible HTLV-1 reservoirs in the body, but we believe that delaying disease progression and diminishing the impact of virus-induced inflammation by boosting a mixed profile of CD8+ T cell responses is possible. Therefore, proper vaccination protocols should aim at not only controlling viral spread but mostly modulating inflammation within the host, which ultimately will decrease the chances of exacerbated inflammation and tissue damage. We revised the text and edited a few paragraphs in the discussion section in order to bring light to this important topic raised by the Reviewer.

In this context, the results of this study will be more convincing if the relationship between infected cells and immune responses in an animal model is presented as a set. Otherwise, the authors should discuss more about the mode of Tax-CTL activation by different Tax epitopes and realistic infected cells in carriers/patients.

Response: We agree with the Reviewer's suggestion regarding the experiments with HTLV carriers and, as advised, we have included CD8 T cell activation results on HTLV-1-infected carriers after an in vitro challenge with C-Tax. As the signal of CD8+ T cell activation for single peptides is so low and similar to the control background, we have focused on the C-terminal portion of Tax to confirm the mixed pro-inflammatory and modulatory profile proposed by the previous in silico and in vitro assays. C-terminal Tax has displayed a robust performance in activating CD8 T cells by inducing both IFN-gamma and IL-4, which confirms our present results on the C-terminal portion of Tax. It is important to note that this data supports the main message of our study, which is the epitope mapping of Tax using the novel Pepperprint technology and the findings of non-Tax11-19 peptides. The data on C-terminal tax is included in the new Figure 7 (pages 13 and 14, lines 409-436) and new discussion on it is now included in the revised version of the manuscript (page 16, lines 570-582).

<Specific Comments>

  1. Line 102: The page is not found.

Response: We apologize for the missing information. A new functional link is provided in the revised version of the manuscript, page 3, lines 108-109.

  1. Line 147; Why was anti-mouse IgG/Cy3 secondary antibody used for human HLA-A02:01 microarray?

Response: Our microarray is based on an antibody (mouse IgG) fused to HLA-A*02:01 as a primary reagent that binds to 9 amino acid peptides adsorbed in the glass slide. To identify which peptides bind to HLA-A*02:01 and at what magnitude, we used a secondary reagent, the anti-mouse IgG conjugated to the fluorescence Cy3 that binds to the mouse IgG part and is then read out by the microarray reader. This is well described now in page 3, lines 133-138. 

  1. Line 375-379: Although Tax plays a critical role in viral replication in the early stage of infection, its role in the maintenance of chronic infection is limited. At which stage the Tax vaccination would be applied?

Response: We believe that proper vaccination protocols should aim at not only controlling viral spread but mostly modulating inflammation within the host during chronic infection, which ultimately will decrease the chances of exacerbated inflammation and tissue damage at late stages of infection This is now better discussed in the revised version of the manuscript as a whole.

  1. Line 390: Tax also induces apoptosis.

Response: This is a correct observation. A paragraph counterbalancing the pro-apoptotic effect of Tax is now included in page 14, lines 467-469 of the discussion section.

  1. Line 439-447: Any relationship between HLA haplotype and HTLV-1 related diseases?

Response: In fact, there is an association of HLA haplotype and HTLV-1 related diseases. This is now included in page 15, lines 485-491 of the revised version of the manuscript. 

  1. Line 481-483: The interesting result that the C-terminal side of Tax is important as an IL-4 inducer from CTLs should be discussed in terms of the relationship between infected cells and the immune system during the early and chronic infection period of HTLV-1.

Response: This is a very important suggestion. The data on C-terminal tax is included in the new Figure 7 (pages 13 and 14) and new discussion on it is now included in the revised version of the manuscript (page 16, lines 565-570).

  1. Line 409-419: These are the main findings of the current study. Most of part of the discussion is a good review of Tax CTL, but more discussion to evaluate the current study is necessary.

Response: As requested by Reviewer #1, we have included additional discussion on our results for the differential HLA-A*02:01 reactivity observed by peptides as compared to Tax11-19. Additional analyses on peptides that outperformed Tax11-19 were also included in the revised figure 3 (former figure 4) in order to clarify this query. The new results are on pages 8 and 9, lines 293-334. 

At last, we appreciate Reviewer#1 for the thorough and careful review of our manuscript. We have worked out very carefully to provide appropriate changes to the queries proposed and we believe that the manuscript has improved in quality after the consideration made by the Reviewer. We would like to thank the Reviewer for revising the manuscript. Please, let us know if any additional changes are yet required.

Best Regards,

The corresponding author

Reviewer 2 Report

In this study, authors profile immunodominant HLA-A2 peptides related to the N-, Central, and C-terminal of the HTLV-1 Tax viral oncoprotein using in silico analysis and a peptide microarray. HLA-A*02:01 has been well established as a protective haplotype for HTLV-1 with the immunodominance of Tax (11-19) epitope, which is restricted to A2. Like other bioinformatics studies published before on HTLV-1 epitopes, the current study also identifies several  Tax A2-reactive peptides and postulates that conserved leucine (L) residues at positions 2 and 9 are important for peptide/HLA-A2 affinity. They also show that the Tax c-terminal region provides many IL-4-inducing epitopes. 

To design a therapeutic vaccine for HTLV-1, it is essential to understand the difference in immune response that leaves 95% of individuals asymptomatic and the remaining 5% with chronic disease. However, this study has some conceptual and technical drawbacks that diminishes the impact as detailed below: 

1.     It is now well known that in silico design of epitopes differ from the natural processing and presentation of peptides. In this regard, a relatively recent study (Vaccine, 2018) provided a list of neo epitopes directly presented by HTLV-1-infected CD4 T cells; identified and characterized by the immunoproteomics. Interestingly, the Tax (11-19) epitope was not identified as the high affinity binder but rather exhibited low binding. This and other studies on HTLV-1 epitope discoveries are either not mentioned or considered in proper context. Therefore, the rationale of the study is poorly developed and experimental validation of in silico epitopes is missing thereby minimizing overall impact. 

2.     Since A2 is a protective haplotype prevalent among non-endemic regions; this kind of in silico mining would be useful if it involved more relevant haplotypes, such as HLA-A24 and HLA-DR, which are prevalent in endemic regions of Japan and South America, respectively. 

3.     This study is conducted in Brazil where there is no rarity of HTLV-1-infected patients with HAM/TSP or ATL. The authors can easily validate their finding experientially with clinical samples. 

4.     Utilizing a prediction algorithm to narrow down potential immunogenic epitopes is a useful tool, however, the paper does not narrow down on a select few candidates worth exploring. 

5.     Moreover, there is no evaluation (in vitro or in vivo) of newly identified epitopes, at least a few from high affinity binders should be tested for the CD8 T-cell activation and cytolytic potential. 

6.     Figure 1 is redundant, just an illustration of all combined figures. A simple graphical abstract of the study flow chart would be more effective. 

7.     In Figure 3, the authors should define reactivity, is this simply the peptides’ ability to bind to the HLA-A2 grove. Also, Figure 3B and 3C seem redundant in relation to Figure 3A. 

8.     Figure 4B and 4C, shows no new data.  Highlighting specific epitopes that outperformed Tax (11-19) would make a stronger figure. 

Considering these comments, the study needs to go through the major revision. 

Author Response

Dear Reviewer #2,

Please, find below the responses to your queries. We have modified the text in the revised version of the manuscript and modifications were marked in yellow.

REVIEWER#2 

Comments and Suggestions for Authors

In this study, authors profile immunodominant HLA-A2 peptides related to the N-, Central, and C-terminal of the HTLV-1 Tax viral oncoprotein using in silico analysis and a peptide microarray. HLA-A*02:01 has been well established as a protective haplotype for HTLV-1 with the immunodominance of Tax (11-19) epitope, which is restricted to A2. Like other bioinformatics studies published before on HTLV-1 epitopes, the current study also identifies several Tax A2-reactive peptides and postulates that conserved leucine (L) residues at positions 2 and 9 are important for peptide/HLA-A2 affinity. They also show that the Tax c-terminal region provides many IL-4-inducing epitopes. 

Response: We appreciate Reviewer#2 for this very accurate summary of our manuscript, which clearly shows that our main message was well understood and conveyed.

To design a therapeutic vaccine for HTLV-1, it is essential to understand the difference in immune response that leaves 95% of individuals asymptomatic and the remaining 5% with chronic disease. However, this study has some conceptual and technical drawbacks that diminishes the impact as detailed below: 

  1.     It is now well known that in silico design of epitopes differ from the natural processing and presentation of peptides. In this regard, a relatively recent study (Vaccine, 2018) provided a list of neo epitopes directly presented by HTLV-1-infected CD4 T cells; identified and characterized by the immunoproteomics. Interestingly, the Tax (11-19) epitope was not identified as the high affinity binder but rather exhibited low binding.

Response: We understand the comment made by Reviewer #2 and we apologize for missing this important report on the immunoproteomics of HLA-A2 restricted HTLV-1 peptides. In fact, this is a very elegant study by Dr. Jain’s laboratory showing novel epitopes able to bind HLA-A2 and HLA-A24. In agreement to our findings, Tax(11-19) did not perform well as a high binder in the immunoproteomics in a similar way that this same peptide performed in our study when low HLA-A2 concentration was employed in the microarray. Therefore, the results of Dr. Jain’s lab confirm our hypothesis that Tax (11-19) is not a high binder under physiological conditions but rather when MHC class I is in very high expression. In order to solve the query proposed by the Reviewer, this discussion as well as the proper citation to Vaccine, 2018 (page 16, line 519-525) is now included in the revised discussion of the manuscript.

This and other studies on HTLV-1 epitope discoveries are either not mentioned or considered in proper context. Therefore, the rationale of the study is poorly developed and experimental validation of in silico epitopes is missing thereby minimizing overall impact. 

Response: We apologize for the missing references to the studies regarding HTLV-1 epitope discoveries. We have worked out to include the proper context to the manuscript as well as references in the epitope discovery. Regarding the validation of the in vitro assay, new results focused on assessing CD8+ T cell activation upon C terminal Tax stimuli are now included in the revised version of the manuscript. The data on C-terminal tax is included in the new Figure 7 (pages 13 and 14, lines 409-436) and new discussion on it is now included in the revised version of the manuscript.

  1.     Since A2 is a protective haplotype prevalent among non-endemic regions; this kind of in silico mining would be useful if it involved more relevant haplotypes, such as HLA-A24 and HLA-DR, which are prevalent in endemic regions of Japan and South America, respectively. 

Response: We appreciate the very important suggestion proposed by Reviewer #2. As requested, we performed the in silico HLA-A24 and HLA-DR epitope prediction of higher binders that outperformed Tax (11-19) in the microarray. The results were included next to the HLA-A2 epitope prediction in the revised figure 3 (page 8). Regarding HLA-DR, the in silico analysis for several alleles of this haplotype did not identified any strong binder (data not shown) for the 9 amino acid Tax peptides. This is now discussed in the revised version of the manuscript.

  1.     This study is conducted in Brazil where there is no rarity of HTLV-1-infected patients with HAM/TSP or ATL. The authors can easily validate their finding experientially with clinical samples. 

Response: This is an accurate comment made by Reviewer#2. In order to solve the query proposed by Reviewer #2, we have included the preliminary data based on HLA-A2+ HTLV-1 patients which is now included in the new figure 7 (page 13-14). 

Brazil is an endemic area for HTLV and our group has had access to plenty of patient samples in the past. However, due to the COVID-19 pandemic, HTLV carriers have been difficult to track. But still, we were able to find samples from asymptomatic cases, none from HAM/TSP. In regard to ATL, this is extremely rare amongst infected patients from Brazil as compared to other parts of the world, so we do not have access to ATL samples in our region.

  1.     Utilizing a prediction algorithm to narrow down potential immunogenic epitopes is a useful tool, however, the paper does not narrow down on a select few candidates worth exploring. 

Response: We understand the Reviewer's critique and we agree with the suggestion. We have now narrowed down to the top 10 peptides that exhibited higher binding and were able to outperform Tax (11-19). The results are shown in the new figure 3 of the revised manuscript. 

  1.     Moreover, there is no evaluation (in vitro or in vivo) of newly identified epitopes, at least a few from high affinity binders should be tested for the CD8 T-cell activation and cytolytic potential. 

Response: We thank the Reviewer for this comment. New figure 7 of the current version of the manuscript includes new data on CD8 T-cell activation using a C-terminal portion of Tax that we produced in the laboratory. We have focused on the C-terminal portion of Tax to confirm the mixed pro-inflammatory and modulatory profile proposed by the previous in silico and in vitro assays. We believe that this mixed profile is essential to modulate chronic HTLV-1 infection. C-terminal Tax has displayed a robust performance in activating CD8+ T cells by inducing both IFN-gamma and IL-4, which confirms our present results on the C-terminal portion of Tax. It is important to note that this data supports the main message of our study, which is the epitope mapping of Tax using the novel Pepperprint technology and the findings of non-Tax11-19 peptides.  

  1.     Figure 1 is redundant, just an illustration of all combined figures. A simple graphical abstract of the study flow chart would be more effective. 

Response: We acknowledged this comment. We have moved Figure 1 as supplementary figure 1. The other figures were renumbered accordingly.

  1.     In Figure 3, the authors should define reactivity, is this simply the peptides’ ability to bind to the HLA-A2 grove. Also, Figure 3B and 3C seem redundant in relation to Figure 3A. 

Response: We thank Reviewer #2 for this query. We define reactivity as the ability of peptides to bind to HLA-A2, which is proportional to the fluorescence obtained in the microarray system. To clarify this important point, this explanation is now included in the revised section of the results for figure 3. In regards to the Fig 3 panels, Fig 3A shows the dose-dependence of peptides for each Tax region and Fig 3B and 3C show statistical differences amongst N Terminal and C terminal regions, so we would like to kindly ask to keep figure panels 3B and 3C in the figure. This figure now corresponds to the new figure 2 of the revised version of the manuscript, page 7. 

  1.     Figure 4B and 4C, shows no new data.  Highlighting specific epitopes that outperformed Tax (11-19) would make a stronger figure. 

Response: We thank Reviewer#2 for this suggestion. As advised, we have excluded the panels in Fig 4B and 4C. We have added a graph showing the frequency of peptides as well as a table highlighting specific epitopes that outperformed the canonic Tax (11-19) to make a stronger figure. This figure is represented as the new figure 3 in the revised version of the manuscript. New data on the tax epitopes of figure 3 are found in page 8 and 10, lines 289-333. 

Considering these comments, the study needs to go through the major revision. 

Response: We appreciate Reviewer#2 for the thorough and careful review of our manuscript. We have worked out very carefully to provide appropriate changes to the queries proposed and we believe that the manuscript has improved in quality after the consideration made by the Reviewer. We would like to thank the Reviewer for revising the manuscript. Please, let us know if any additional changes are yet required.

Best Regards,

The corresponding author

Round 2

Reviewer 1 Report

No more comments for authors.